# Diverse and Accurate Image Description Using a Variational Auto-Encoder with an Additive Gaussian Encoding Space

**Liwei Wang**          **Alexander G. Schwing**          **Svetlana Lazebnik**
`{lwang97, aschwing, slazebni}@illinois.edu`
University of Illinois at Urbana-Champaign

## Abstract

This paper explores image caption generation using conditional variational auto-encoders (CVAEs). Standard CVAEs with a fixed Gaussian prior yield descriptions with too little variability. Instead, we propose two models that explicitly structure the latent space around $K$ components corresponding to different types of image content, and combine components to create priors for images that contain multiple types of content simultaneously (e.g., several kinds of objects). Our first model uses a Gaussian Mixture model (GMM) prior, while the second one defines a novel Additive Gaussian (AG) prior that linearly combines component means. We show that both models produce captions that are more diverse and more accurate than a strong LSTM baseline or a "vanilla" CVAE with a fixed Gaussian prior, with AG-CVAE showing particular promise.

## 1   Introduction

Automatic image captioning [9, 11, 18–20, 24] is a challenging open-ended conditional generation task. State-of-the-art captioning techniques [23, 32, 36, 1] are based on recurrent neural nets with long-short term memory (LSTM) units [13], which take as input a feature representation of a provided image, and are trained to maximize the likelihood of reference human descriptions. Such methods are good at producing relatively short, generic captions that roughly fit the image content, but they are unsuited for sampling multiple diverse candidate captions given the image. The ability to generate such candidates is valuable because captioning is profoundly ambiguous: not only can the same image be described in many different ways, but also, images can be hard to interpret even for humans, let alone machines relying on imperfect visual features. In short, we would like the posterior distribution of captions given the image, as estimated by our model, to accurately capture both the open-ended nature of language and any uncertainty about what is depicted in the image.

Achieving more diverse image description is a major theme in several recent works [6, 14, 27, 31, 35]. Deep generative models are a natural fit for this goal, and to date, Generative Adversarial Models (GANs) have attracted the most attention. Dai *et al*. [6] proposed jointly learning a generator to produce descriptions and an evaluator to assess how well a description fits the image. Shetty *et al*. [27] changed the training objective of the generator from reproducing ground-truth captions to generating captions that are indistinguishable from those produced by humans.

In this paper, we also explore a generative model for image description, but unlike the GAN-style training of [6, 27], we adopt the conditional variational auto-encoder (CVAE) formalism [17, 29]. Our starting point is the work of Jain *et al*. [14], who trained a "vanilla" CVAE to generate questions given images. At training time, given an image and a sentence, the CVAE encoder samples a latent $z$ vector from a Gaussian distribution in the encoding space whose parameters (mean and variance) come from a Gaussian prior with zero mean and unit variance. This $z$ vector is then fed into a decoder that uses it, together with the features of the input image, to generate a question. The encoder and the decoder are jointly trained to maximize (an upper bound on) the likelihood of the reference questions

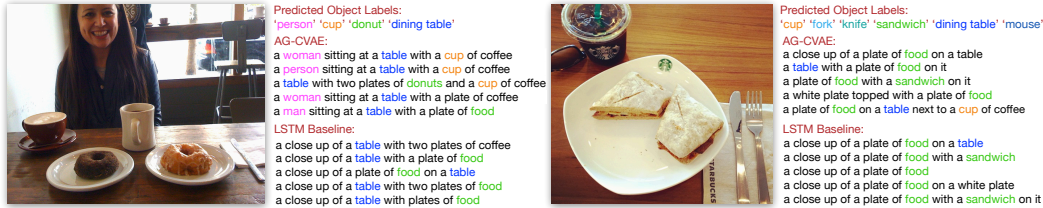

Figure 1: Example output of our proposed AG-CVAE approach compared to an LSTM baseline (see Section 4 for details). For each method, we show top five sentences following consensus re-ranking [10]. The captions produced by our method are both more diverse and more accurate.

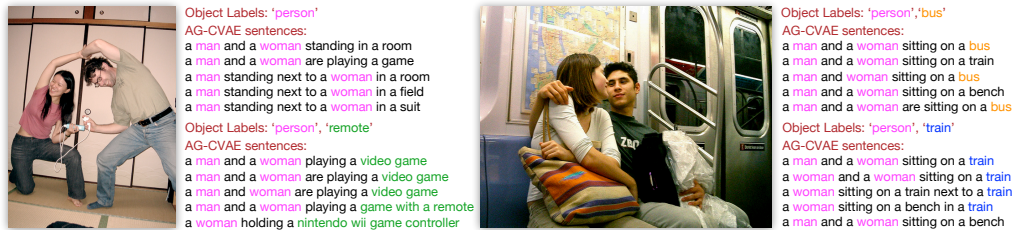

Figure 2: Illustration of how our additive latent space structure controls the image description process. Modifying the object labels changes the weight vectors associated with semantic components in the latent space. In turn, this shifts the mean from which the $z$ vectors are drawn and modifies the resulting descriptions in an intuitive way.

given the images. At test time, the decoder is seeded with an image feature and different $z$ samples, so that multiple $z$'s result in multiple questions.

While Jain *et al.* [14] obtained promising question generation performance with the above CVAE model equipped with a fixed Gaussian prior, for the task of image captioning, we observed a tendency for the learned conditional posteriors to collapse to a single mode, yielding little diversity in candidate captions sampled given an image. To improve the behavior of the CVAE, we propose using a set of $K$ Gaussian priors in the latent $z$ space with different means and standard deviations, corresponding to different "modes" or types of image content. For concreteness, we identify these modes with specific object categories, such as 'dog' or 'cat.' If 'dog' and 'cat' are detected in an image, we would like to encourage the generated captions to capture both of them.

Starting with the idea of multiple Gaussian priors, we propose two different ways of structuring the latent $z$ space. The first is to represent the distribution of $z$ vectors using a Gaussian Mixture model (GMM). Due to the intractability of Gaussian mixtures in the VAE framework, we also introduce a novel Additive Gaussian (AG) prior that directly adds multiple semantic aspects in the $z$ space. If an image contains several objects or aspects, each corresponding to means $\mu_k$ in the latent space, then we require the mean of the encoder distribution to be close to a weighted linear combination of the respective means. Our CVAE formulation with this additive Gaussian prior (AG-CVAE) is able to model a richer, more flexible encoding space, resulting in more diverse and accurate captions, as illustrated in Figure 1. As an additional advantage, the additive prior gives us an interpretable mechanism for controlling the captions based on the image content, as shown in Figure 2. Experiments of Section 4 will show that both GMM-CVAE and AG-CVAE outperform LSTMs and "vanilla" CVAE baselines on the challenging MSCOCO dataset [5], with AG-CVAE showing marginally higher accuracy and by far the best diversity and controllability.

## 2 Background

Our proposed framework for image captioning extends the standard variational auto-encoder [17] and its conditional variant [29]. We briefly set up the necessary background here.

**Variational auto-encoder (VAE):** Given samples $x$ from a dataset, VAEs aim at modeling the data likelihood $p(x)$. To this end, VAEs assume that the data points $x$ cluster around a low-dimensional manifold parameterized by embeddings or encodings $z$. To obtain the sample $x$ corresponding to an embedding $z$, we employ the *decoder* $p(x|z)$ which is often based on deep nets. Since the decoder's posterior $p(z|x)$ is not tractably computable we approximate it with a distribution $q(z|x)$ which is

referred to as the *encoder*. Taking together all those ingredients, VAEs are based on the identity

$$\log p(x) - D_{\mathrm{KL}}[q(z|x), p(z|x)] = \mathbb{E}_{q(z|x)}[\log p(x|z)] - D_{\mathrm{KL}}[q(z|x), p(z)], \tag{1}$$

which relates the likelihood $p(x)$ and the conditional $p(z|x)$. It is hard to compute the KL-divergence $D_{\mathrm{KL}}[q(z|x), p(z|x)]$ because the posterior $p(z|x)$ is not readily available from the decoder distribution $p(x|z)$ if we use deep nets. However, by choosing an encoder distribution $q(z|x)$ with sufficient capacity, we can assume that the non-negative KL-divergence $D_{\mathrm{KL}}[q(z|x), p(z|x)]$ is small. Thus, we know that the right-hand-side is a lower bound on the log-likelihood $\log p(x)$, which can be maximized w.r.t. both encoder and decoder parameters.

**Conditional variational auto-encoders (CVAE):** In tasks like image captioning, we are interested in modeling the *conditional* distribution $p(x|c)$, where $x$ are the desired descriptions and $c$ is some representation of content of the input image. The VAE identity can be straightforwardly extended by conditioning both the encoder and decoder distributions on $c$. Training of the encoder and decoder proceeds by maximizing the lower bound on the conditional data-log-likelihood $p(x|c)$, *i.e.*,

$$\log p_\theta(x|c) \geq \mathbb{E}_{q_\phi(z|x,c)}[\log p_\theta(x|z, c)] - D_{\mathrm{KL}}[q_\phi(z|x, c), p(z|c)], \tag{2}$$

where $\theta$ and $\phi$, the parameters for the decoder distribution $p_\theta(x|z, c)$ and the encoder distribution $q_\phi(z|x, c)$ respectively. In practice, the following stochastic objective is typically used:

$$\max_{\theta, \phi} \frac{1}{N} \sum_{i=1}^{N} \log p_\theta(x^i|z^i, c^i) - D_{\mathrm{KL}}[q_\phi(z|x, c), p(z|c)], \quad \text{s.t.} \ \forall i \ z^i \sim q_\phi(z|x, c).$$

It approximates the expectation $\mathbb{E}_{q_\phi(z|x,c)}[\log p_\theta(x|z, c)]$ using $N$ samples $z^i$ drawn from the approximate posterior $q_\phi(z|x, c)$ (typically, just a single sample is used). Backpropagation through the encoder that produces samples $z^i$ is achieved via the reparameterization trick [17], which is applicable if we restrict the encoder distribution $q_\phi(z|x, c)$ to be, *e.g.*, a Gaussian with mean and standard deviation output by a deep net.

## 3 Gaussian Mixture Prior and Additive Gaussian Prior

Our key observation is that the behavior of the trained CVAE crucially depends on the choice of the prior $p(z|c)$. The prior determines how the learned latent space is structured, because the KL-divergence term in Eq. (2) encourages $q_\phi(z|x, c)$, the encoder distribution over $z$ given a particular description $x$ and image content $c$, to be close to this prior distribution. In the vanilla CVAE formulation, such as the one adopted in [14], the prior is not dependent on $c$ and is fixed to a zero-mean unit-variance Gaussian. While this choice is the most computationally convenient, our experiments in Sec. 4 will demonstrate that for the task of image captioning, the resulting model has poor diversity and worse accuracy than the standard maximum-likelihood-trained LSTM. Clearly, the prior has to change based on the content of the image. However, because of the need to efficiently compute the KL-divergence in closed form, it still needs to have a simple structure, ideally a Gaussian or a mixture of Gaussians.

Motivated by the above considerations, we encourage the latent $z$ space to have a multi-modal structure composed of $K$ modes or clusters, each corresponding to different types of image content. Given an image $I$, we assume that we can obtain a distribution $c(I) = (c_1(I), \ldots, c_K(I))$, where the entries $c_k$ are nonnegative and sum to one. In our current work, for concreteness, we identify these with a set of object categories that can be reliably detected automatically, such as 'car,' 'person,' or 'cat.' The MSCOCO dataset, on which we conduct our experiments, has direct supervision for 80 such categories. Note, however, our formulation is general and can be applied to other definitions of modes or clusters, including latent topics automatically obtained in an unsupervised fashion.

**GMM-CVAE**: We can model $p(z|c)$ as a Gaussian mixture with weights $c_k$ and components with means $\mu_k$ and standard deviations $\sigma_k$:

$$p(z|c) = \sum_{k=1}^{K} c_k \mathcal{N}\left(z \,|\mu_k, \ \sigma_k^2 \mathrm{I}\right), \tag{3}$$

where $c_k$ is defined as the weights above and $\mu_k$ represents the mean vector of the $k$-th component. In practice, for all components, we use the same standard deviation $\sigma$.

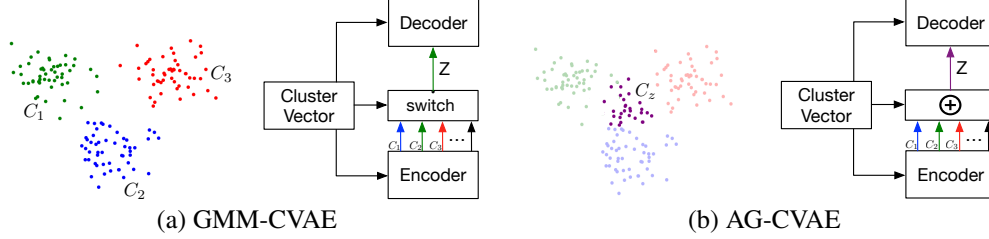

(a) GMM-CVAE               (b) AG-CVAE

Figure 3: Overview of GMM-CVAE and AG-CVAE models. To sample $z$ vectors given an image, GMM-CVAE (a) switches from one cluster center to another, while AG-CVAE (b) encourages the embedding $z$ for an image to be close to the average of its objects' means.

It is not directly tractable to optimize Eq. (2) with the above GMM prior. We therefore approximate the KL divergence stochastically [12]. In each step during training, we first draw a discrete component $k$ according to the cluster probability $c(I)$, and then sample $z$ from the resulting Gaussian component. Then we have

$$
\begin{aligned}
D_{\mathrm{KL}}[q_\phi(z|x,c_k), p(z|c_k)] &= \log\left(\frac{\sigma_k}{\sigma_\phi}\right) + \frac{1}{2\sigma^2}\mathbb{E}_{q_\phi(z|x,c_k)}\left[\|z - \mu_k\|_2^2\right] - \frac{1}{2} \\
&= \log\left(\frac{\sigma_k}{\sigma_\phi}\right) + \frac{\sigma_\phi^2 + \|\mu_\phi - \mu_k\|_2^2}{2\sigma_k^2} - \frac{1}{2}, \quad \forall k \;\; c_k \sim c(I).
\end{aligned}
\tag{4}
$$

We plug the above KL term into Eq. (2) to obtain an objective function, which we optimize w.r.t. the encoder and decoder parameters $\phi$ and $\theta$ using stochastic gradient descent (SGD). In principle, the prior parameters $\mu_k$ and $\sigma_k$ can also be trained, but we obtained good results by keeping them fixed (the means are drawn randomly and all standard deviations are set to the same constant, as will be further explained in Section 4).

At test time, in order to generate a description given an image $I$, we first sample a component index $k$ from $c(I)$, and then sample $z$ from the corresponding component distribution. One limitation of this procedure is that, if an image contains multiple objects, each individual description is still conditioned on just a single object.

**AG-CVAE**: We would like to structure the $z$ space in a way that can directly reflect object co-occurrence. To this end, we propose a simple novel conditioning mechanism with an additive Gaussian prior. If an image contains several objects with weights $c_k$, each corresponding to means $\mu_k$ in the latent space, we want the mean of the encoder distribution to be close to the linear combination of the respective means with the same weights:

$$
p(z|c) = \mathcal{N}\left(z \,\middle|\, \sum_{k=1}^{K} c_k\mu_k, \; \sigma^2\mathrm{I}\right),
\tag{5}
$$

where $\sigma^2\mathrm{I}$ is a spherical covariance matrix with $\sigma^2 = \sum_{k=1}^{K} c_k^2\sigma_k^2$. Figure 3 illustrates the difference between this AG-CVAE model and the GMM-CVAE model introduced above.

In order to train the AG-CVAE model using the objective of Eq. (2), we need to compute the KL-divergence $D_{\mathrm{KL}}[q_\phi(z|x,c), p(z|c)]$ where $q_\phi(z|x,c) = \mathcal{N}(z \,|\, \mu_\phi(x,c), \sigma_\phi^2(x,c)\mathrm{I})$ and the prior $p(z|c)$ is given by Eq. (5). Its analytic expression can be derived to be

$$
\begin{aligned}
D_{\mathrm{KL}}[q_\phi(z|x,c), p(z|c)] &= \log\left(\frac{\sigma}{\sigma_\phi}\right) + \frac{1}{2\sigma^2}\mathbb{E}_{q_\phi}\left[\left\|z - \sum_{k=1}^{K} c_k\mu_k\right\|^2\right] - \frac{1}{2} \\
&= \log\left(\frac{\sigma}{\sigma_\phi}\right) + \frac{\sigma_\phi^2 + \|\mu_\phi - \sum_{k=1}^{K} c_k\mu_k\|^2}{2\sigma^2} - \frac{1}{2}.
\end{aligned}
$$

We plug the above KL-divergence term into Eq. (2) to obtain the stochastic objective function for training the encoder and decoder parameters. We initialize the mean and variance parameters $\mu_k$ and $\sigma_k$ in the same way as for GMM-CVAE and keep them fixed throughout training.

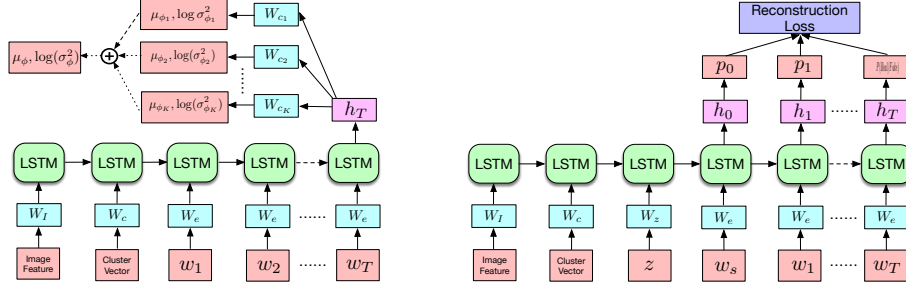

Figure 4: Illustration of our encoder (left) and decoder (right). See text for details.

Next, we need to specify our architectures for the encoder and decoder, which are shown in Fig. 4.

**The encoder** uses an LSTM to map an image $I$, its vector $c(I)$, and a caption into a point in the latent space. More specifically, the LSTM receives the image feature in the first step, the cluster vector in the second step, and then the caption word by word. The hidden state $h_T$ after the last step is transformed into $K$ mean vectors, $\mu_{\phi k}$, and $K$ log variances, $\log \sigma_{\phi k}^2$, using a linear layer for each. For AG-CVAE, the $\mu_{\phi k}$ and $\sigma_{\phi k}^2$ are then summed with weights $c_k$ and $c_k^2$ respectively to generate the desired $\mu_\phi$ and $\sigma_\phi^2$ encoder outputs. Note that the encoder is used at training time only, and the input cluster vectors are produced from ground truth object annotations.

**The decoder** uses a different LSTM that receives as input first the image feature, then the cluster vector, then a $z$ vector sampled from the conditional distribution of Eq. (5). Next, it receives a 'start' symbol and proceeds to output a sentence word by word until it produces an 'end' symbol. During training, its $c(I)$ inputs are derived from the ground truth, same as for the encoder, and the log-loss is used to encourage reconstruction of the provided ground-truth caption. At test time, ground truth object vectors are not available, so we rely on automatic object detection, as explained in Section 4.

## 4 Experiments

### 4.1 Implementation Details

We test our methods on the MSCOCO dataset [5], which is the largest "clean" image captioning dataset available to date. The current (2014) release contains 82,783 training and 40,504 validation images with five reference captions each, but many captioning works re-partition this data to enlarge the training set. We follow the train/val/test split released by [23]. It allocates $118,287$ images for training, $4,000$ for validation, and $1,000$ for testing.

**Features.** As image features, we use 4,096-dimensional activations from the VGG-16 network [28]. The cluster or object vectors $c(I)$ are 80-dimensional, corresponding to the 80 MSCOCO object categories. At training time, $c(I)$ consist of binary indicators corresponding to ground truth object labels, rescaled to sum to one. For example, an image with labels 'person,' 'car,' and 'dog' results in a cluster vector with weights of $1/3$ for the corresponding objects and zeros elsewhere. For test images $I$, $c(I)$ are obtained automatically through object detection. We train a Faster R-CNN detector [26] for the MSCOCO categories using our train/val split by fine-tuning the VGG-16 net [28]. At test time, we use a threshold of 0.5 on the per-class confidence scores output by this detector to determine whether the image contains a given object (*i.e.*, all the weights are once again equal).

**Baselines.** Our **LSTM** baseline is obtained by deleting the $z$ vector input from the decoder architecture shown in Fig. 4. This gives a strong baseline comparable to NeuralTalk2 [1] or Google Show and Tell [33]. To generate different candidate sentences using the LSTM, we use beam search with a width of 10. Our second baseline is given by the "vanilla" **CVAE** with a fixed Gaussian prior following [14]. For completeness, we report the performance of our method as well as all baselines both with and without the cluster vector input $c(I)$.

**Parameter settings and training.** For all the LSTMs, we use a one-hot encoding with vocabulary size of 11,488, which is the number of words in the training set. This input gets projected into a word embedding layer of dimension 256, and the LSTM hidden space dimension is 512. We found that the same LSTM settings worked well for all models. For our three models (CVAE, GMM-CVAE, and AG-CVAE), we use a dimension of 150 for the $z$ space. We wanted it to be at least equal to the number of categories to make sure that each $z$ vector corresponds to a unique set of cluster weights. The means $\mu_k$ of clusters for GMM-CVAE and AG-CVAE are randomly initialized on the unit ball

| | obj | #z | std | beam | B4 | B3 | B2 | B1 | C | R | M | S |
|---|---|---|---|---|---|---|---|---|---|---|---|---|
| LSTM | - | - | - | 10 | 0.413 | 0.515 | 0.643 | 0.790 | 1.157 | 0.597 | 0.285 | 0.218 |
| | ✓ | - | - | 10 | 0.428 | 0.529 | 0.654 | 0.797 | 1.202 | 0.607 | 0.290 | 0.223 |
| CVAE | - | 20 | 0.1 | - | 0.261 | 0.381 | 0.538 | 0.742 | 0.860 | 0.531 | 0.246 | 0.184 |
| | ✓ | 20 | 2 | - | 0.312 | 0.421 | 0.565 | 0.733 | 0.910 | 0.541 | 0.244 | 0.176 |
| GMM-CVAE | - | 20 | 0.1 | - | 0.371 | 0.481 | 0.619 | 0.778 | 1.080 | 0.582 | 0.274 | 0.209 |
| | ✓ | 20 | 2 | - | 0.423 | 0.533 | 0.666 | 0.813 | 1.216 | 0.617 | 0.298 | 0.233 |
| | ✓ | 20 | 2 | 2 | 0.449 | 0.553 | 0.680 | 0.821 | 1.251 | 0.624 | 0.299 | 0.232 |
| | ✓ | 100 | 2 | - | 0.494 | 0.597 | 0.719 | 0.856 | 1.378 | 0.659 | 0.325 | 0.261 |
| | ✓ | 100 | 2 | 2 | 0.527 | 0.625 | 0.740 | 0.865 | 1.430 | 0.670 | 0.329 | 0.263 |
| AG-CVAE | - | 20 | 0.1 | - | 0.431 | 0.537 | 0.668 | 0.814 | 1.230 | 0.622 | 0.300 | 0.235 |
| | ✓ | 20 | 2 | - | 0.451 | 0.557 | 0.686 | 0.829 | 1.259 | 0.630 | 0.305 | 0.243 |
| | ✓ | 20 | 2 | 2 | 0.471 | 0.573 | 0.698 | 0.834 | 1.308 | 0.638 | 0.309 | 0.244 |
| | ✓ | 100 | 2 | - | 0.532 | 0.631 | 0.749 | 0.876 | 1.478 | 0.682 | 0.342 | 0.278 |
| | ✓ | 100 | 2 | 2 | 0.557 | 0.654 | 0.767 | 0.883 | 1.517 | 0.690 | 0.345 | 0.277 |

Table 1: Oracle (upper bound) performance according to each metric. Obj indicates whether the object (cluster) vector is used; #z is the number of $z$ samples; std is the test-time standard deviation; beam is the beam width if beam search is used. For the caption quality metrics, C is short for Cider, R for ROUGE, M for METEOR, S for SPICE.

| | obj | #z | std | beam | B4 | B3 | B2 | B1 | C | R | M | S |
|---|---|---|---|---|---|---|---|---|---|---|---|---|
| LSTM | - | - | - | 10 | 0.286 | 0.388 | 0.529 | 0.702 | 0.915 | 0.510 | 0.235 | 0.165 |
| | ✓ | - | - | 10 | 0.292 | 0.395 | 0.536 | 0.711 | 0.947 | 0.516 | 0.238 | 0.170 |
| CVAE | - | 20 | 0.1 | - | 0.245 | 0.347 | 0.495 | 0.674 | 0.775 | 0.491 | 0.217 | 0.147 |
| | ✓ | 20 | 2 | - | 0.265 | 0.372 | 0.521 | 0.698 | 0.834 | 0.506 | 0.225 | 0.158 |
| GMM-CVAE | - | 20 | 0.1 | - | 0.271 | 0.376 | 0.522 | 0.702 | 0.890 | 0.507 | 0.231 | 0.166 |
| | ✓ | 20 | 2 | - | 0.278 | 0.388 | 0.538 | 0.718 | 0.932 | 0.516 | 0.238 | 0.170 |
| | ✓ | 20 | 2 | 2 | 0.289 | 0.394 | 0.538 | 0.715 | 0.941 | 0.513 | 0.235 | 0.169 |
| | ✓ | 100 | 2 | - | 0.292 | 0.402 | 0.552 | 0.728 | 0.972 | 0.520 | 0.241 | 0.174 |
| | ✓ | 100 | 2 | 2 | 0.307 | 0.413 | 0.557 | 0.729 | 0.986 | 0.525 | 0.242 | 0.177 |
| AG-CVAE | - | 20 | 0.1 | - | 0.287 | 0.394 | 0.540 | 0.715 | 0.942 | 0.518 | 0.238 | 0.168 |
| | ✓ | 20 | 2 | - | 0.286 | 0.391 | 0.537 | 0.716 | 0.953 | 0.517 | 0.239 | 0.172 |
| | ✓ | 20 | 2 | 2 | 0.299 | 0.402 | 0.544 | 0.716 | 0.963 | 0.518 | 0.237 | 0.173 |
| | ✓ | 100 | 2 | - | 0.301 | 0.410 | 0.557 | 0.732 | 0.991 | 0.527 | 0.243 | 0.177 |
| | ✓ | 100 | 2 | 2 | 0.311 | 0.417 | 0.559 | 0.732 | 1.001 | 0.528 | 0.245 | 0.179 |

Table 2: Consensus re-ranking using CIDEr. See caption of Table 1 for legend.

and are not changed throughout training. The standard deviations $\sigma_k$ are set to 0.1 at training time and tuned on the validation set at test time (the values used for our results are reported in the tables). All networks are trained with SGD with a learning rate that is 0.01 for the first 5 epochs, and is reduced by half every 5 epochs. On average all models converge within 50 epochs.

## 4.2 Results

A big part of the motivation for generating diverse candidate captions is the prospect of being able to re-rank them using some discriminative method. Because the performance of any re-ranking method is upper-bounded by the quality of the best candidate caption in the set, we will first evaluate different methods assuming an oracle that can choose the best sentence among all the candidates. Next, for a more realistic evaluation, we will use a consensus re-ranking approach [10] to automatically select a single top candidate per image. Finally, we will assess the diversity of the generated captions using uniqueness and novelty metrics.

**Oracle evaluation.** Table 1 reports caption evaluation metrics in the oracle setting, *i.e.*, taking the maximum of each relevant metric over all the candidates. We compare caption quality using five metrics: BLEU [25], METEOR [7], CIDEr [30], SPICE [2], and ROUGE [21]. These are calculated using the MSCOCO caption evaluation tool [5] augmented by the author of SPICE [2]. For the LSTM baseline, we report the scores attained among 10 candidates generated using beam search (as suggested in [23]). For CVAE, GMM-CVAE and AG-CVAE, we sample a fixed number of $z$ vectors from the corresponding prior distributions (the numbers of samples are given in the table).

The high-level trend is that "vanilla" CVAE falls short even of the LSTM baseline, while the upper-bound performance for GMM-CVAE and AG-CVAE considerably exceeds that of the LSTM given

| | obj | #z | std | beam size | % unique per image | % novel sentences |
|---|---|---|---|---|---|---|
| LSTM | ✓ | - | - | 10 | - | 0.656 |
| CVAE | ✓ | 20 | 2 | - | 0.118 | 0.820 |
| GMM-CVAE | ✓ | 20 | 2 | - | 0.594 | 0.809 |
| | ✓ | 20 | 2 | 2 | 0.539 | 0.716 |
| | ✓ | 100 | 2 | - | 0.376 | 0.767 |
| | ✓ | 100 | 2 | 2 | 0.326 | 0.688 |
| AG-CVAE | ✓ | 20 | 2 | - | 0.764 | 0.795 |
| | ✓ | 20 | 2 | 2 | 0.698 | 0.707 |
| | ✓ | 100 | 2 | - | 0.550 | 0.745 |
| | ✓ | 100 | 2 | 2 | 0.474 | 0.667 |

Table 3: Diversity evaluation. For each method, we report the percentage of unique candidates generated per image by sampling different numbers of $z$ vectors. We also report the percentage of novel sentences (i.e., sentences not seen in the training set) out of (at most) top 10 sentences following consensus re-ranking. It should be noted that for CVAE, there are 2,466 novel sentences out of 3,006. For GMM-CVAE and AG-CVAE, we get roughly 6,200-7,800 novel sentences.

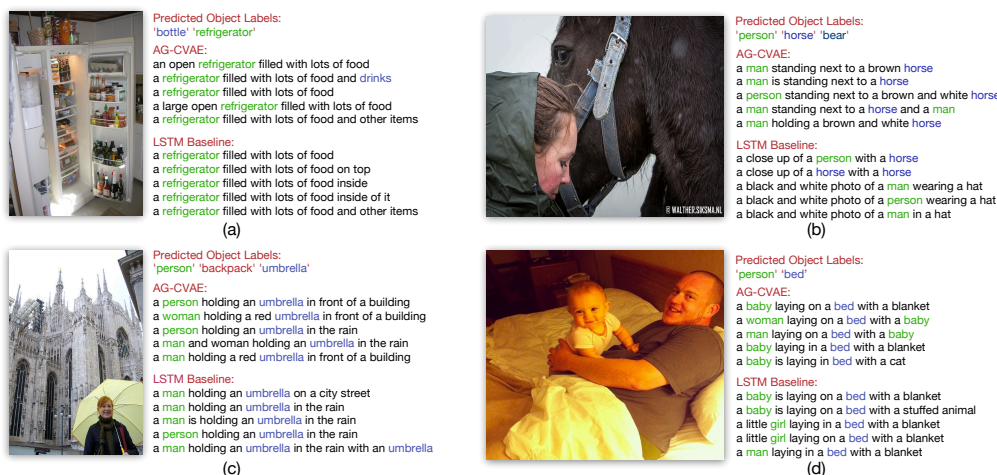

Figure 5: Comparison of captions produced by our AG-CVAE method and the LSTM baseline. For each method, top five captions following consensus re-ranking are shown.

the right choice of standard deviation and a large enough number of $z$ samples. AG-CVAE obtains the highest upper bound. A big advantage of the CVAE variants over the LSTM is that they can be easily used to generate more candidate sentences simply by increasing the number of $z$ samples, while the only way to do so for the LSTM is to increase the beam width, which is computationally prohibitive.

In more detail, the top two lines of Table 1 compare performance of the LSTM with and without the additional object (cluster) vector input, and show that it does not make a dramatic difference. That is, improving over the LSTM baseline is not just a matter of adding stronger conditioning information as input. Similarly, for CVAE, GMM-CVAE, and AG-CVAE, using the object vector as additional conditioning information in the encoder and decoder can increase accuracy somewhat, but does not account for all the improvements that we see. One thing we noticed about the models without the object vector is that they are more sensitive to the standard deviation parameter and require more careful tuning (to demonstrate this, the table includes results for several values of $\sigma$ for the CVAE models).

**Consensus re-ranking evaluation.** For a more realistic evaluation we next compare the same models after consensus re-ranking [10, 23]. Specifically, for a given test image, we first find its nearest neighbors in the training set in the cross-modal embedding space learned by a two-branch network proposed in [34]. Then we take all the ground-truth reference captions of those neighbors and calculate the consensus re-ranking scores between them and the candidate captions. For this, we use the CIDEr metric, based on the observation of [22, 30] that it can give more human-consistent evaluations than BLEU.

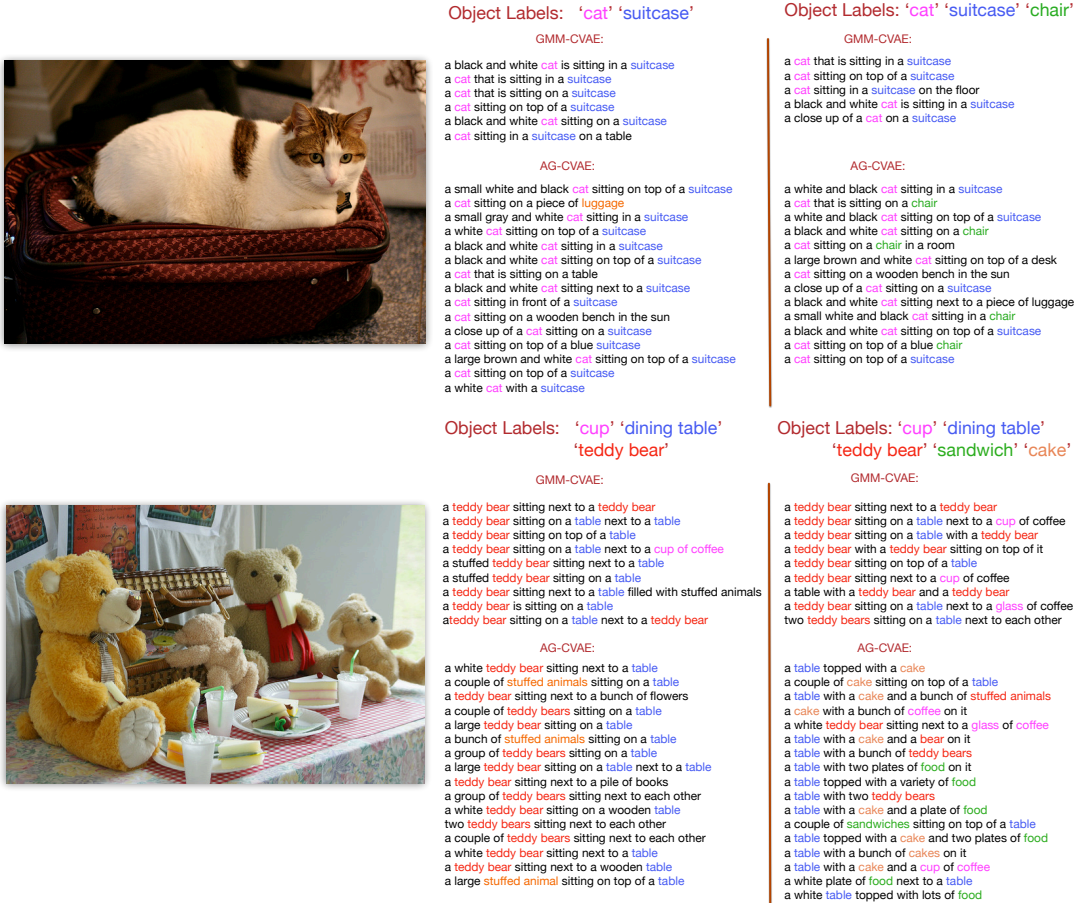

Figure 6: Comparison of captions produced by GMM-CVAE and AG-CVAE for two different versions of input object vectors for the same images. For both models, we draw 20 $z$ samples and show the resulting unique captions.

Table 2 shows the evaluation based on the single top-ranked sentence for each test image. While the re-ranked performance cannot get near the upper bounds of Table 1, the numbers follow a similar trend, with GMM-CVAE and AG-CVAE achieving better performance than the baselines in almost all metrics. It should also be noted that, while it is not our goal to outperform the state of the art in absolute terms, our performance is actually better than some of the best methods to date [23, 37], although [37] was trained on a different split. AG-CVAE tends to get slightly higher numbers than GMM-CVAE, although the advantage is smaller than for the upper-bound results in Table 1. One of the most important take-aways for us is that there is still a big gap between upper-bound and re-ranking performance and that improving re-ranking of candidate sentences is an important future direction.

**Diversity evaluation**. To compare the generative capabilities of our different methods we report two indicative numbers in Table 3. One is the average percentage of unique captions in the set of candidates generated for each image. This number is only meaningful for the CVAE models, where we sample candidates by drawing different $z$ samples, and multiple $z$'s can result in the same caption. For LSTM, the candidates are obtained using beam search and are by definition distinct. From Table 3, we observe that CVAE has very little diversity, GMM-CVAE is much better, but AG-CVAE has the decisive advantage.

Similarly to [27], we also report the percentage of all generated sentences for the test set that have not been seen in the training set. It only really makes sense to assess novelty for sentences that are plausible, so we compute this percentage based on (at most) top 10 sentences per image after consensus re-ranking. Based on the novelty ratio, CVAE does well. However, since it generates fewer distinct candidates per image, the absolute numbers of novel sentences are much lower than for GMM-CVAE and AG-CVAE (see table caption for details).

**Qualitative results.** Figure 5 compares captions generated by AG-CVAE and the LSTM baseline on four example images. The AG-CVAE captions tend to exhibit a more diverse sentence structure with a wider variety of nouns and verbs used to describe the same image. Often this yields captions that are more accurate ('open refrigerator' vs. 'refrigerator' in (a)) and better reflective of the cardinality and types of entities in the image (in (b), our captions mention both the person and the horse while the LSTM tends to mention only one). Even when AG-CVAE does not manage to generate any correct candidates, as in (d), it still gets the right number of people in some candidates. A shortcoming of AG-CVAE is that detected objects frequently end up omitted from the candidate sentences if the LSTM language model cannot accommodate them ('bear' in (b) and 'backpack' in (c)). On the one hand, this shows that the capacity of the LSTM decoder to generate combinatorially complex sentences is still limited, but on the other hand, it provides robustness against false positive detections.

**Controllable sentence generation.** Figure 6 illustrates how the output of our GMM-CVAE and AG-CVAE models changes when we change the input object vectors in an attempt to control the generation process. Consistent with Table 3, we observe that for the same number of $z$ samples, AG-CVAE produces more unique candidates than GMM-CVAE. Further, AG-CVAE is more flexible than GMM-CVAE and more responsive to the content of the object vectors. For the first image showing a cat, when we add the additional object label 'chair,' AG-CVAE is able to generate some captions mentioning a chair, but GMM-CVAE is not. Similarly, in the second example, when we add the concepts of 'sandwich' and 'cake,' only AG-CVAE can generate some sentences that capture them. Still, the controllability of AG-CVAE leaves something to be desired, since, as observed above, it has trouble mentioning more than two or three objects in the same sentence, especially in unusual combinations.

## 5 Discussion

Our experiments have shown that both our proposed GMM-CVAE and AG-CVAE approaches generate image captions that are more diverse and more accurate than standard LSTM baselines. While GMM-CVAE and AG-CVAE have very similar bottom-line accuracies according to Table 2, AG-CVAE has a clear edge in terms of diversity (unique captions per image) and controllability, both quantitatively (Table 3) and qualitatively (Figure 6).

**Related work.** To date, CVAEs have been used for image question generation [14], but as far as we know, our work is the first to apply them to captioning. In [8], a mixture of Gaussian prior is used in CVAEs for colorization. Their approach is essentially similar to our GMM-CVAE, though it is based on mixture density networks [4] and uses a different approximation scheme during training.

Our CVAE formulation has some advantages over the CGAN approach adopted by other recent works aimed at the same general goals [6, 27]. GANs do not expose control over the structure of the latent space, while our additive prior results in an interpretable way to control the sampling process. GANs are also notoriously tricky to train, in particular for discrete sampling problems like sentence generation (Dai *et al.* [6] have to resort to reinforcement learning and Shetty *et al.* [27] to an approximate Gumbel sampler [15]). Our CVAE training is much more straightforward.

While we represent the $z$ space as a simple vector space with multiple modes, it is possible to impose on it a more general graphical model structure [16], though this incurs a much greater level of complexity. Finally, from the viewpoint of inference, our work is also related to general approaches to diverse structured prediction, which focus on extracting multiple modes from a single energy function [3]. This is a hard problem necessitating sophisticated approximations, and we prefer to circumvent it by cheaply generating a large number of diverse and plausible candidates, so that "good enough" ones can be identified using simple re-ranking mechanisms.

**Future work.** We would like to investigate more general formulations for the conditioning information $c(I)$, not necessarily relying on object labels whose supervisory information must be provided separately from the sentences. These can be obtained, for example, by automatically clustering nouns or noun phrases extracted from reference sentences, or even clustering vector representations of entire sentences. We are also interested in other tasks, such as question generation, where the cluster vectors can represent the question type ('what is,' 'where is,' 'how many,' *etc.*) as well as the image content. Control of the output by modifying the $c$ vector would in this case be particularly natural.

**Acknowledgments:** This material is based upon work supported in part by the National Science Foundation under Grants No. 1563727 and 1718221, and by the Sloan Foundation. We would like to thank Jian Peng and Yang Liu for helpful discussions.

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
