[Reviews · NeurIPS 2017]

Reviewer 1



This paper investigated the task of image-conditioned caption generation using deep generative models. Compared to existing methods with pure LSTM pipeline, the proposed approach augments the representation with an additional data dependent latent variable. This paper formulated the problem under variational auto-encoder (VAE) framework by maximizing the variational lowerbound as objective during training. A data-dependent additive Gaussian prior was introduced to address the issue of limited representation power when applying VAEs to caption generation. Empirical results demonstrate the proposed method is able to generate diverse and accurate sentences compared to pure LSTM baseline. == Qualitative Assessment == I like the motivation that adding stochastic latent variable to the caption generation framework. Augmenting the prior of VAE is not a novel contribution but I see the novelty applied to caption generation task. Performance-wise, the proposed AG-CVAE achieved more accurate performance compared to both LSTM baseline and other CVAE baselines (see Table 2). The paper also analyzed the diversity of the generated captions in comparison with the pure LSTM based approach (see Figure 5). Overall, the paper is generally well-written with sufficient experimental details. Considering the additive Gaussian prior as major contribution, the current version does not seem to be very convincing to me. I am happy to raise score if my concerns can be addressed in the rebuttal. * Any strong evidence showing the advantages of AG-CVAE over CVAE/GMM-CVAE? The improvements in Table 2 are not very significant. For qualitative results (Figure 5 and other figures in supplementary materials), side-by-side comparisons between AG-CVAE and CVAE/GMM-CVAE are missing. It is not crystal clear to me whether AG-CVAE actually adds more representation power compared to CVAE/GMM-CVAE. Please comment on this in the rebuttal and include such comparisons in the final version of the paper (or supplementary materials). * Diversity evaluation: it is not clear why does AG-CVAE perform worse than CVAE. Also, there is no explanation about performance gap from different variations of AG-CVAE. Since CVAE is a stochastic generative model, I wonder whether top 10 sentences are sufficient for diversity evaluations? The results will be much more convincing if the authors provide a curve (y-axis is the diversity measure and x-axis is the number of sentences). Additional comments: * Pre-defined means of clusters for GMM-CVAE and AG-CVAE (Line 183) It is not surprising that the authors failed to obtain better results when means of clusters u_k are free variables. In principle, it is possible to learn duplicate representations without any constraints (e.g., sparsity or orthogonality) on u_k or c_k. I would encourage the authors to explore this direction a bit in the future. Hopefully, learnable data-dependent prior can boost the performance to some extent.

Reviewer 2



In general, I find the paper to be fairly well-written and the idea to be both quite intuitive and well motivated. However I do have a couple of questions regarding several technical decisions made in the paper. * For the encoder (Fig. 4), is it really necessary to explicitly generate K posterior mean/variances separately before taking the weighted average to combine them into one? Or, compared with having the encoder network directly generate the posterior mean/variances, how much gain would the current architecture have? * Given the fact that ultimately we only need the decoder to do conditional caption generation (i.e. there is no **actual** need for doing posterior inference at all), it seems to me that VAE might be an overkill here. Why not directly learn the decoder via maximum-likelihood? You can actually still sample z from p(z|c) and then marginalize it to approximately maximize p(x|c) = \sum_{z_i} p(z_i|c)p(x|z_i, c). This should be an even stronger LSTM baseline with z still present. * For GMM-CVAE, how did you compute the KL-divergence? * During training, the standard deviations \sigma_k in the prior essentially controls the balance between KL-divergence (as the regularization) and reconstruction error (as the loss). I'm therefore interested to know whether the authors have tried different values as they did during test time. I raise this issue also because to me it is more reasonable to stick with the same \sigma_k both for training and testing. Typo * L.36: "maximize (an upper bound on) the likelihood" should have been "lower bound" I think.

Reviewer 3



This paper presents and tests two variants of a prior on the VAE latent space for image captioning. Specifically they both employ a Gaussian mixture model prior on the latent apace, and then a better performing variant whereby the clusters are encourage to represent objects in a given image --- i.e. any given image will be generated via a linear combination of samples from clusters corresponding to the objects it contains. Detailed comments: * The paper is well written and appears to be well tested against previous baselines. While the idea is straightforward, and perhaps not a standalone contribution by itself the testing appears to be reasonably conclusive, and would appear to yield state of the art results (when checked against the online MSCOCO baseline dataset). I am curious as to why the CGAN was not also benchmarked, however (there are now techniques for sentence generating GAN models via e.g. the Gumbel-Softmax trick)? * It would be interesting to combine the GMM latent space model with a DPP prior over the cluster centers, to further encourage diversity in representation, a hierarchical latent space might also be an interesting direction to explore, although I am strongly of the opinion that the performance of these models (particularly on small datasets) is entirely dependent on how close the implicit prior encoded by the choice of mapping and latent structure is to the true data generating distribution, in which case it may be that the AG-CVAE is close to optimal.